# Exosomal Liquid Biopsy in Prostate Cancer: A Systematic Review of Biomarkers for Diagnosis, Prognosis, and Treatment Response

**DOI:** 10.3390/ijms26020802

**Published:** 2025-01-18

**Authors:** Yameen Hamid, Rukhshana Dina Rabbani, Rakkan Afsara, Samarea Nowrin, Aruni Ghose, Vasileios Papadopoulos, Konstantinos Sirlantzis, Saak V. Ovsepian, Stergios Boussios

**Affiliations:** 1The University of Edinburgh, Edinburgh EH8 9YL, UK; muhammad.hamid3@nhs.net; 2Department of Acute Medicine, Medway NHS Foundation Trust, Gillingham ME7 5NY, UK; 3Department of Medical Oncology, Medway NHS Foundation Trust, Gillingham ME7 5NY, UK; rukhshana.rabbani@nhs.net (R.D.R.); aruni.ghose1@gmail.com (A.G.); 4Department of Medical Oncology, Evercare Hospital, Dhaka 1205, Bangladesh; tushiafsararakkan125@gmail.com; 5Department of Clinical Oncology, Maidstone and Tunbridge Wells NHS Trust, Maidstone ME16 9QQ, UK; samarea.nowrin@nhs.net; 6Department of Urology, Kent and Canterbury Hospital, Canterbury CT1 3NG, UK; vpapadoster@gmail.com; 7School of Engineering, Technology and Design, Canterbury Christ Church University, Canterbury CT1 1QU, UK; konstantinos.sirlantzis@canterbury.ac.uk; 8Faculty of Engineering and Science, University of Greenwich London, Chatham Maritime, Kent ME4 4TB, UK; s.v.ovsepian@greenwich.ac.uk; 9Faculty of Medicine, Tbilisi State University, Tbilisi 0177, Georgia; 10Faculty of Medicine, Health, and Social Care, Canterbury Christ Church University, Canterbury CT2 7PB, UK; 11School of Cancer and Pharmaceutical Sciences, Faculty of Life Sciences and Medicine, King’s College London, Strand, London WC2R 2LS, UK; 12Kent and Medway Medical School, University of Kent, Canterbury CT2 7LX, UK; 13AELIA Organisation, 9th km Thessaloniki—Thermi, 57001 Thessaloniki, Greece

**Keywords:** prostate cancer, liquid biopsy, exosomes, biomarkers, miRNA, prognosis, response

## Abstract

Prostate cancer, a leading cause of cancer-related mortality among men, often presents challenges in accurate diagnosis and effective monitoring. This systematic review explores the potential of exosomal biomolecules as noninvasive biomarkers for the diagnosis, prognosis, and treatment response of prostate cancer. A thorough systematic literature search through online public databases (Medline via PubMed, Scopus, and Web of science) using structured search terms and screening using predefined eligibility criteria resulted in 137 studies that we analyzed in this systematic review. We evaluated the findings from these clinical studies, revealing that the load of exosomes in the blood and urine of prostate cancer patients, which includes microRNAs (miRNAs), proteins, and lipids, demonstrates disease-specific changes. It also shows that some exosomal markers can differentiate between malignant and benign hyperplasia of the prostate, predict disease aggressiveness, and monitor treatment efficacy. Notably, miRNA emerged as the most frequently studied biomolecule, demonstrating superior diagnostic potential compared to traditional methods like prostate-specific antigen (PSA) testing. The analysis also highlights the pressing need for a standardised analytic approach through multi-centre studies to validate the full potential of exosomal biomarkers for the diagnosis and monitoring of prostate cancer.

## 1. Introduction

Prostate cancer is the second most commonly diagnosed cancer and the fifth leading cause of cancer death among men worldwide, with an estimated 1.5 million new cases and 397,000 deaths worldwide [1]. A range of genetic, hereditary, and environmental factors contribute to an increased risk of developing prostate cancer, including advancing age, a family history of prostate cancer and African ethnicity.

Screening of prostate cancer is performed globally by digital rectal examination (DRE), the prostate-specific antigen (PSA) blood test, and transrectal ultrasound (TRUS)-guided biopsy [2]. Elevated PSA in serum is of limited specificity to prostate cancer, detecting only one-fifth of patients, raising concerns over the accuracy of testing and diagnosis [3]. DRE is commonly practised in clinical settings for screening, though it falls short in early diagnosis [4]. Tissue biopsy is challenging and comes with a risk of infection, bleeding, and pain associated with the procedures [2]. Recent large-scale genomic sequencing efforts have provided new insight into the genetic landscape of prostate cancer. Mutations in DNA damage repair genes are present in 19% of primary prostate cancer cases and nearly 23% of metastatic castration-resistant prostate cancer (mCRPC) cases, leading to compromised genomic integrity [5,6]. Regarding molecular screening, the IMPACT study confirmed *BReast CAncer* gene 1 and 2 (*BRCA1/2*) genotyping could aid in detecting aggressive variants of prostate cancer, recommending surveillance for the carriers of this genotype [7,8]. Therapeutically, agents targeting DNA damage repair pathways are being explored as standalone treatments or in combination with therapies that induce DNA damage, with ongoing clinical trials assessing their effectiveness in prostate cancer management [9].

The diagnosis of prostate cancer relies on the microscopic analysis of prostate tissue obtained through transrectal ultrasonography (TRUS) in a grid-like pattern. Based on the cellular architecture and appearance, the biopsy results are evaluated against the primary Gleason group grade for the predominant histological and malignancy patterns. Serum PSA variant tests help estimate the prostate cancer risk in patients with prior negative biopsies. Advanced imaging techniques have also been incorporated into diagnostic protocols, and, most notably, magnetic resonance imaging (MRI), which utilises specialised sequences alongside T2-weighted imaging [10]. The sensitivity and specificity of MRI for detecting prostate cancer are reported to be 89% and 73%, respectively [11]. Patients are categorised as having a low, intermediate, or high risk based on their Gleason scores, PSA levels, and clinical stage [12,13]. The interest in molecular or functional imaging using positron emission tomography (PET) has grown, with several radiotracers showing promise in prostate cancer detection. Of these, three—C-choline, 18F-fluciclovine, and 18F-sodium fluoride—have received FDA approval [14]. PET-CT and PET-MRI have shown comparative advantages over existing imaging methods, particularly for detecting regional lymph node metastases and in patients with low PSA levels. Recently, multiple studies have illustrated the application of artificial intelligence (AI) in PET/CT for prostate cancer, showcasing its utility in various clinical measures such as lesion detection, delineation, and outcome prediction [15].

Current treatment protocols for localised prostate cancer include active surveillance, radical prostatectomy, and ablative radiotherapy. Active surveillance involves regular PSA testing, physical exams, and prostate biopsies, either individually or in combination, to monitor disease progression [16]. For patients with more advanced disease, such as those presenting with PSA levels above ten ng/mL or palpable nodules detected via DRE, surgery and radiation remain effective treatment options. The latter has advanced with the introduction of intensity-modulated radiation therapy (IMRT), which has primarily replaced 3D-conformal radiation therapy. Stereotactic ablative radiotherapy (SABR) is an innovative and promising treatment for men with oligometastatic prostate cancer. SABR has demonstrated safety and efficacy and is appealing compared to other ablative techniques due to its noninvasive nature, providing an outpatient procedure that can be administered using a standard linear accelerator [17,18]. For metastatic prostate cancer, androgen deprivation therapy (ADT) is the mainstay of first-line treatment [19,20]. Newer agents have emerged that target the androgen axis; abiraterone acetate inhibits androgen biosynthesis, while enzalutamide, darolutamide, and apalutamide disrupt androgen-receptor signalling [19,20,21,22,23,24,25]. Furthermore, biallelic inactivation of Cyclin Dependent Kinase 12 (CDK12) is linked to a distinct genome instability phenotype characterised by CDK12-specific focal tandem duplications, which can result in the altered expression of oncogenic drivers like Cyclin D1 (CCND1) and CDK4. These advances suggest the potential susceptibility of CDK12-mutated tumours to CDK4/6 inhibitors [26].

Exosomes are extracellular vesicles secreted by cells, ranging from 40 to 100 nm in diameter and constructed of a spherical lipid bilayer [27,28]. They are formed as intraluminal vesicles by the inward folding of the endosomal membrane during the formation of multivesicular endosomes (MVEs). After reaching maturity, these MVEs fuse with the cell membrane, releasing the exosomes [29,30]. The exosome content that has been characterised in great detail includes nucleic acids, lipids, proteins, transcription factor receptors, cytokines, and various metabolites [31]. Exosomes play a crucial role in physiological processes such as intercellular communication, angiogenesis, inflammation, metabolic regulation, and others, with their dysfunction contributing to metabolic, cardiovascular, and neurodegenerative disease and cancers [32,33,34,35].

There is a growing trend toward favouring liquid biopsies over tissue biopsies due to their less invasive nature and systemic approach. In addition to circulating neoplastic cells and DNA fragments found in bodily fluids, exosomes in liquid biopsies provide a wealth of information about the molecular composition of tumours. Urinary liquid biopsy presents an appealing and promising method for detecting prostate cancer. Beyond specific urine biomarkers, potential serum biomarkers that could drive the precision medicine revolution include androgen receptor variants, markers of bone metabolism, neuroendocrine indicators, and metabolite biomarkers [36]. Within this context, the extraction and analysis of exosomes from the liquid biopsy samples (i.e., blood, urine, and semen) of prostate cancer patients have shown significant advantages as a source of potential biomarkers [37,38]. We identified that the currently available reviews on similar topics are narrative reviews, address all types of liquid biopsy contents in prostate cancer [39,40], discuss the role of liquid biopsy in all types of cancer [41,42], focus on particular exosomal contents [43], or address the role of exosomes in particular subgroups of prostate cancer patients [44]. Thus, we designed this systematic review protocol to evaluate and integrate comprehensive evidence supporting the utility of exosomal biomolecules in liquid biopsy for the diagnosis, prognosis, and treatment response of prostate cancer.

## 2. Methods

### 2.1. Search Strategy

The protocol of this systematic review was registered with PROSPERO (registration number: CRD42024522627). A systematic search was performed in the Medline database through the PubMed, Web of Science Core Collection, and Scopus online databases. The articles were searched from inception till September 2024. The search strategy included keywords such as ‘prostate cancer’, ‘exosome’, ‘diagnosis’, ‘prognosis’, ‘therapy response’, and ‘role’, following the PICO (population, intervention, comparison, and outcome) framework. We formulated four different comprehensive search terms using the keywords, which were (prostate cancer) AND (exosome) AND (diagnosis), (prostate cancer) AND (exosome) AND (prognosis), (prostate cancer) AND (exosome) AND (therapy response) and (prostate cancer) AND (exosome) AND (role). In PubMed and Web of Science Core Collection, the search was performed in the ‘all field’ section, and in the Scopus database, the search was performed in the ‘title, abstract, keywords’ section. A total of 2170 records were found initially from all three databases. After eliminating duplicated records using Rayyan [45], a web-based software for systematic reviews, 973 articles were selected for title and abstract screening.

### 2.2. Study Selection

The following were inclusion criteria for this systematic review: (i) cohort and case–control studies including patient samples; (ii) sample source was liquid biopsy (i.e., blood or urine); and (iii) effect was measured with inferential statistics (i.e., Receiver Operating Characteristic (ROC) or analysis of variance (ANOVA)). Records were excluded based on the following: (i) in vitro studies, bioinformatic studies, review articles; (ii) samples including tumour or tissue biopsy only; (iii) studies including other cancers apart from prostate cancer; and (iv) articles not written in English.

### 2.3. Data Extraction and Synthesis

Four independent reviewers screened the articles and performed the full-text assessment. Initial title and abstract screening was performed using ASReview software LAB v1.5, and 160 articles were put forward for the full-text evaluation [46]. At least two independent reviewers reviewed each article. Any conflict was resolved by consensus between the reviewers. After a thorough review and assessment for eligibility, 137 articles were selected based on eligibility criteria for data extraction and synthesis (Figure 1). Extracted data include name and type of exosomal component, role in disease diagnosis, prognosis or therapy response, measurement of effect (expression level, *p*-value, sensitivity, specificity, area under ROC curve, etc.), type of study, source of sample, and sample size. Extracted data were compiled in a master spreadsheet, and further tabulation was carried out based on the data categories in this article. The Preferred Reporting Items for Systematic Reviews and Meta-Analyses (PRISMA) checklist 2020 was used to report the systematic review [47].

### 2.4. Quality Assessment

The articles’ quality and risk of bias assessments were performed using the Newcastle–Ottawa quality assessment scale (NOS), as it is suitable for assessing non-randomised studies [48]. NOS assesses the quality of an article in three domains: selection, comparability, and outcome assessment. Each article was assigned a score from 0 (zero) to 9 (nine). Any article with a score more than 7 (seven) was considered a high-quality article for this systematic review.

## 3. Results

### 3.1. General Features of the Included Articles

In this systematic review, we included 137 articles, of which 91 were case–control studies, and 46 were cohort studies. Most of the case–control studies included healthy individuals as control groups. However, 36 studies included patients with benign prostate hyperplasia (BPH) as control groups, and 13 articles used multiple control groups in their studies. The majority of the cohort studies were retrospective, and only two studies were prospective cohorts. The earliest paper we included is from the year 2009, and the latest is from the year of writing, 2024. Most of the articles were published between 2018 and 2024, and most of the papers included in this systematic review were from 2021 (Figure 2).

In total, 17,419 patients with prostate cancer of different stages were studied in the included articles. Among them, 13,303 were enrolled in cohort studies and 4116 were enrolled in case–control studies. As control groups, 1701 healthy individuals and 3638 patients with BPH were enrolled. Two studies enrolled prostatitis patients as the control group [49,50], and two enrolled healthy individuals with raised PSA levels [51,52].

We included studies that used liquid biopsy samples to measure the effect of exosomal components. The most frequent samples were blood, either plasma or serum (73 articles), and urine (64 articles), from which exosomes were isolated. There were only two studies that explored semen, one study of prostatic fluid, and one study examined saliva. Three of the studies used multiple liquid biopsy samples.

The exosomal contents that were investigated included proteins, RNAs, lipids, metabolites, and fragments of DNA. Different categories of RNA were investigated, among which microRNAs (miRNA) were mostly observed, comprising 49 articles (Figure 3). The rest of the types were messenger RNAs (mRNAs), long noncoding RNAs (lncRNA), small noncoding RNAs (sncRNA), PIWI-interacting RNAs (piRNA), and circular RNAs (circRNA). mRNA were measured to quantify particular gene expression. The most abundant types of proteins that were investigated were cell surface proteins and cancer-related proteins (oncoproteins), followed by hormones, cytokines, and enzymes.

We explored the articles that investigated the role of different exosomal contents in the diagnosis, prognosis, and treatment response of prostate cancer. Most of the studies examined the possibility of utilising exosomal contents as a diagnostic biomarker (94 articles). The prognosis of the disease was described in terms of the overall survival (OS), risk groups, disease classification (i.e., Gleason score or group grade), biochemical recurrence-free survival, and cancer metastasis. Forty-eight of the articles investigated the role of exosomes in disease prognosis, and twenty examined the role of using exosomes as biomarkers to measure treatment response. Twenty-seven studies explored exosomes’ role in multiple domains of prostate cancer (Figure 4).

### 3.2. Exosomal Contents with Diagnostic Value

Most studies evaluating exosomal contents for potential diagnostic biomarkers measured various miRNAs (34 articles). These articles studied 59 different types of miRNA overall (Table 1). The majority reported the overexpression of miRNA; however, a few studies reported the decreased expression of certain miRNAs in prostate cancer patients [53,54,55,56,57]. In most of the reports of diagnostic efficacy with ROC, the area under the ROC curve (AUC) was significantly higher than the gold standard for prostate cancer diagnosis—the analysis of the serum PSA level. One study mentioned that miRNA (miR-145) combined with PSA can infer a better diagnostic value [58].

Two articles studied piRNAs for their role as diagnostic biomarkers [59,60]. Peng et al. examined four novel piRNAs that yielded significant AUCs in the ROC analysis [59]. Merkert et al. studied piRNAs and miRNAs, finding that the AUC was more than 0.7 for most individual biomarkers [60]. At least eight different lncRNAs were analysed in five articles, all of which found that lncRNAs were significantly overexpressed in prostate cancer. In one study, the authors developed a urinary exosomal lncRNA assay incorporating PCA3 and MALAT1 to diagnose prostate cancer and high-grade prostate cancer. This assay achieved a superior AUC in predicting biopsy results, outperforming current clinical parameters [61].

The protein expression, measured by Western blots, enzyme-linked immunosorbent assays (ELISAs), or sodium dodecyl sulfate-polyacrylamide gel electrophoresis (SDS-PAGE), was examined in 30 articles. Most of them measured either cancer-related proteins (i.e., PSMA, PCA3, TMPRSS2:EGR, or PSA) or exosomal cell surface proteins (CD9, CD3, EpCAM, or CD81). Two articles found cytokines as a potent diagnostic biomarker [62,63]. Two articles found some enzymes (Carbonic anhydrase IX and Gamma-glutamyltransferase) to be significantly increased in prostate cancer [64,65].

Sixteen articles studied the expression of multiple genes that had the potential for detecting prostate cancer. Most of these genes are prostate cancer-related genes (i.e., ERG, PCA3, and AR-V7). Two studied the role of ExoDx, a commercial three urinary gene expression (ERG, PCA3, and SPDEF)-based method (Table 2), and validated its diagnostic and prognostic efficacy in larger cohorts [66,67]. Chu et al. found that sex-related steroid hormones (i.e., dehydroepiandrosterone and testosterone) have a significant diagnostic role in prostate cancer [68]. Additionally, one study found elevated urinary exosomal glycoprofile marker, while another report found changes in five different circRNA, which could be utilised as potential diagnostic biomarkers [69,70]. There is a common trend to observe that exosomal biomarkers are generally overexpressed in prostate cancer or associated with high-grade disease or poor therapy response, except a few studies that found the downregulation of the biomarkers detailed in Table 1 and Table 2.

**Table 1 ijms-26-00802-t001:** List of different exosomal biomolecules with diagnostic value.

Year	Authors (et al.)	Exosome Contents	Type of Exosome Content	Detection Method	Sample Source	Sample Size	Change in Expression	Type of Study	Reference
2022	Zhang H	SERPINA3, LRG1, SCGB3A1	Protein	Mass spectrometry	Blood	PC: 20, prostatitis: 20	Increased	Case–control	[50]
2023	Ge Q	PSEP and PSA	Protein	ELISA	Urine	AP: 54, CP: 72, PC: 36	Increased	Case–control	[51]
2021	Matsuzaki K	miR-30b-3p and miR-126-3p	miRNA	Microarray	Urine	PC: 10 biopsy negative raised PSA: 4	Increased	Case–control	[52]
2022	Luedemann C	mir-331-3p and mir-200b	miRNA	qRT-PCR	Saliva	PC: 43, control: 31 (raised PSA)	Reduced	Case–control	[53]
2020	Zhou C	miR-217 and miR-23b-3p	miRNA	Small RNA sequencing	Blood	PC: 10, HC: 10	miR-217: increased, miR-23b-3p: reduced	Case–control	[54]
2023	Cruz-Burgos M	miR-150-5p	miRNA	Microarray	Blood	PC: 20, HC: 15	Reduced	Case–control	[55]
2017	Rodríguez M	miR-196a-5p, miR-501-3p	miRNA	Next-generation sequencing	Urine	PC: 28, HC: 19	Reduced	Case–control	[56]
2024	Evin D	miR-15a, miR-16, miR-19a-3p, and miR-21	miRNA	qRT-PCR	Blood	mCRPC: 51, BPH: 48	Reduced	Case–control	[58]
2017	Xu Y	MiR-145	miRNA	qRT-PCR	Urine	PC: 60, BPH: 37	Increased	Case–control	[59]
2021	Peng Q	novel_pir349843, novel_pir382289, novel_pir158533, and hsa_piR_002468	piRNA	RT-qPCR	Urine	PC: 30, HC: 10	Increased	Case–control	[60]
2021	Markert L	miR-15p, miR-3126-3p, miR-324-5p, miR-150-5p, miR-425-3p, miR-6078, piR_018849, piR_019324	piRNA and miRNA	Next-generation sequencing	Urine	BPH: 25, PC: 28	miR-6078: increased, rest are reduced	Case–control	[61]
2021	Li Y	PCA3, MALAT1, and lncRNA	mRNA and lncRNA	qRT-PCR	Urine	PC: 218 BPH: 347	Increased	Case–control	[62]
2023	Xu F	Interleukin 8	Protein	ELISA	Blood	PC: 32, HC: 10	Increased	Case–control	[63]
2020	Logozzi M	Carbonic anhydrase IX	Protein	ELISA	Blood	PC: 8, HC: 8	Increased	Case–control	[65]
2017	Kawakami K	Gamma-glutamyltransferase	Protein	Western blot	Blood	PC: 31, BPH: 8	Increased	Case–control	[66]
2017	Vermassen T	UGM, PSA	Glycoprofile and protein	Multicapillary carbohydrate electrophoresis and immunoassay	Urine	PC: 85, BPH: 122	Increased	Case–control	[70]
2023	Liu P	Apolipoprotein E, LRG1 and ITIH3, metabolites	Protein	Mass spectrometry	Blood	PC: 30, HC: 15	Increased	Case–control	[71]
2022	Zhai TY	miR-20b-5p	miRNA	RT-qPCR	Prostatic fluid	PC: 10, HC: 27	Increased	Case–control	[72]
2019	Danarto R	miR-21-5p and miR-200c-3p	miRNA	RT-qPCR	Urine	BPH: 20, PC:60	miR-21-5p: increased, miR-200c-3p: reduced	Case–control	[73]
2012	Khan S	Survivin	Protein	ELISA	Blood	PC: 39, BPH: 20, Recurrent PC:8, HC: 16	Increased	Case–control	[74]
2021	Ji J	CDC42, IL32, MAX, NCF2, PDGFA, and SRSF2	mRNA	RNA sequencing	Blood	PC: 141, BPH: 170, HC: 30	Increased	Case–control	[75]
2023	Wang CB	PSMA	Protein	ELISA	Urine	BPH: 194, PC: 80	Increased	Case–control	[76]
2015	Korzeniewski N	miRNA-483-5p	miRNA	Microarray	Urine	HC: 18, PC: 71	Increased	Case–control	[77]
2021	Li Q	lncRNA AY927529	lncRNA	Western blot	Blood	PC: 10, HC: 10	Increased	Case–control	[78]
2021	Kohaar I	PCA3, PCGEM1	mRNA	droplet digital PCR	Urine	N: 271	Increased	Cohort	[79]
2016	Motamedinia P	TMPRSS2:ERG	mRNA	RT-qPCR	Urine	PC: 84, Control: 88	Increased	Case–control	[80]
2016	Bryzgunova OE	miR-19b	miRNA	qRT-PCR	Urine	PC: 14, HC: 20	Reduced	Case–control	[81]
2015	Duijvesz D	CD9, CD63	Protein	Fluorescence immunoassay	Urine	PC: 67, HC: 76	Increased	Case–control	[82]
2009	Lu Q	Delta-catenin	Protein	SDS-PAGE	Urine	PC (Active cancer): 16, Control (Inactive cancer): 15	Increased	Case–control	[83]
2016	Turay D	Apolipoproteins, Pregnancy Zone Protein, Macroglobulins, Keratin, Albumin Precursors, Haptoglobin, Ceruloplasmins, Transferrin, Complement Proteins and Fibronectin	Protein	Mass spectrometry	Blood	PC: 12, HC: 9	Increased	Case–control	[84]
2022	Gan J	ERG, PCA3, PSMA, CK19, and EpCAM	mRNA	RT-qPCR	Urine	PC: 63, HC: 61	Increased	Case–control	[85]
2017	Logozzi M	CD81	Protein	ELISA	Blood	PC: 15, HC: 15	Increased	Case–control	[86]
2017	Worst TS	Claudin 3	Protein	Mass spectrometry	Blood	PC: 58, BPH: 15, HC: 15	Increased	Case–control	[87]
2015	Hata K	NEU3	Protein	ELISA	Blood	PC: 34, HC: 13	Increased	Case–control	[88]
2015	Işın M	lincRNA-p21	lncRNA	qRT-PCR	Urine	PC: 30, BPH control: 49	Increased	Case–control	[89]
2017	Yang JS	Lipids (DAG, TAG, and ChE)	Lipid	Liquid chromatography-tandem mass spectrometry	Urine	PC: 4, HC: 4	Increased	Case–control	[90]
2019	Barceló M	miR-130a-3p, miR-142-3p, miR-142-5p, miR-223-3p,	miRNA	qRT-PCR	Semen	PC: 31, HC: 11, BPH control: 7	Increased	Case–control	[91]
2024	Wang C	miR-320c, miR-944	miRNA	RT-qPCR	Blood	PC: 87, HC: 112	miR-320c: increased, miR-944: reduced	Case–control	[92]
2024	Yu J	RAB5B, WWP1	mRNA	qRT-PCR	Urine	PC: 10, HC: 10	Increased	Case–control	[93]
2021	Dai Y	EpCAM-CD9	Protein	Chemiluminescent immunoassay	Urine	PC: 112, BPH: 55, HC: 26	Reduced	Case–control	[94]
2021	Zabegina L	miR-145, miR-451a, miR-141, and miR-221	miRNA	RT-qPCR	Blood	PC: 55, HC: 30	Increased	Case–control	[95]
2024	Martorana E	STAT3, CyclinD1, ERBB3, ALK, and CD81	Protein	Western blot	Blood	PC: 42, BPH: 12	Increased	Case–control	[96]
2020	McKiernan J	PCA3, ERG and SPDEF (ExoDx)	mRNA	qRT-PCR	Urine	N: 229	Increased	Cohort	[97]
2020	Davey M	FOLH1, HPN, CD24, TMPRSS2-ERG, ITSN1, ANXA3, SLC45A3	mRNA	qRT-PCR	Urine	PC: 28, HC: 28	Increased	Case–control	[98]
2015	Li M	miR375, miR21, miR574	miRNA	qRT-PCR	Blood	PC: 14, HC: 10	Increased	Case–control	[99]
2017	Sanda MG	TMPRSS2-ERG, PCA3	mRNA	qRT-PCR	Urine	Development cohort: 761 Validation cohort: 743	Increased	Cohort	[100]
2019	Gu CY	PSEP	Protein	ELISA	Urine	N: 674	Increased	Cohort	[101]
2017	Endzeliņš E	miR-200c-3p and miR-21-5p	miRNA	RT-qPCR	Blood	PC: 50, BPH: 22	Increased	Case–control	[102]
2021	Kim J	miR-221	miRNA	RT-qPCR	Blood	PC: 39, BPH: 8	Increased	Case–control	[103]
2023	Tutrone R	PCA3, ERG and SPDEF (ExoDx)	mRNA	qRT-PCR	Urine	N: 833	Increased	Cohort	[104]
2017	Foj L	miRNA-21, miR-375, let-7c	miRNA	qRT-PCR	Urine	PC: 60, HC: 10	Increased	Case–control	[105]
2016	Li Z	miRNA-141	miRNA	RT-qPCR	Blood	PC: 20, BPH: 20, HC: 20	Increased	Case–control	[106]
2023	Yazbek Hanna M	GJB1, RPS10, TMPRSS2:ERG, ERG_Exons_4-5, HPN	mRNA	qRT-PCR	Urine	PC: 40, non-cancer: 36	Increased	Case–control	[107]
2015	Øverbye A	TM256 and LAMTOR1	Protein	Mass spectrometry	Urine	PC: 16, HC: 15	Increased	Case–control	[108]
2019	Logozzi M	CD81 and PSA	Protein	Immunocapture-based ELISA	Blood	PC: 80,BPH: 80, HC: 80	Increased	Case–control	[109]
2021	Zhang S	miR-146a-5p,miR-24-3p and miR-93-5p	miRNA	qRT-PCR	Blood	PC: 86, HC: 86	Increased	Case–control	[110]
2022	Holdmann J	miR-532-3p and miR-6749-5p	miRNA	Next-generation sequencing	Urine	PC: 28, BPH: 25	Increased	Case–control	[111]
2020	Worst TS	miR-10a-5p and miR-29b-3p	miRNA	Next-generation sequencing	Blood	PC: 18, BPH: 7	Increased	Case–control	[112]
2017	Skotland T	phosphatidylserine, lactosylceramide	Lipid	High-throughput mass spectrometry	Urine	PC: 15, HC: 13	Increased	Case–control	[113]
2020	Li W	miR-125a-5p and miR-141-5p	miRNA	RT-qPCR	Blood	PC: 38, HC: 19	miR-125a-5p: reduced, miR-141-5p: increased	Case–control	[114]
2014	Dijkstra S	PCA3	mRNA	qRT-PCR	Urine	N: 30	Increased	Cohort	[115]
2017	Khan S	Inhibitors of apoptosisproteins (IAP)	Protein	ELISA	Blood	PC: 72, HC: 10	Increased	Case–control	[116]
2020	Konoshenko MY	miR-19b, miR-30e, miR-31, miR-92a, miR-125, miR-200, miR-205, and miR-660	miRNA	RT-qPCR	Urine	PC: 10, BPH: 8, HC: 11	Increased	Case–control	[117]
2018	Wang YH	SAP30L-AS1 and SChLAP1	lncRNA	qRT-PCR	Blood	PC: 34, BPH: 46, HC: 30	Increased	Case–control	[118]
2024	Jiang S	FOXA1, PCA3, and KLK3	mRNA	RT-qPCR	Urine	Training cohort: 234 Validation cohort:101	Increased	Cohort	[119]
2022	Jin X	AMACR (a-Methylacyl-CoA racemase)	Protein	ELISA	Urine	PC: 87, BPH: 185	Increased	Case–control	[120]
2019	Woo HK	AR-V7, AR-FL	mRNA	droplet digital PCR	Urine	CRPC: 14, hormone-sensitive PC: 22, HC: 11	Increased	Case–control	[121]
2022	Kretschmer A	ERG, PCA3 and SPDEF (ExoDx)	mRNA	qRT-PCR	Urine	N = 109	Increased	Cohort	[122]
2019	Li P	PSA, PCA3	mRNA	qRT-PCR	Urine	PC: 20, HC: 15	Increased	Case–control	[123]
2014	Neeb A	AGR2	mRNA	qRT-PCR	Urine	PC: 24, BPH: 15	Increased	Case–control	[124]
2015	Samsonov R	miR-574-3p, miR-141-5p and miR-21-5p	miRNA	RT-qPCR	Urine	PC: 35, HC: 35	Increased	Case–control	[125]

PC, prostate cancer; AP, acute prostatitis; CP, chronic prostatitis; HC, healthy control; ELISA, enzyme-linked immunosorbent assay; qRT-PCR, quantitative real-time polymerase chain reaction; RT-qPCR, reverse transcription quantitative polymerase chain reaction; SDS-PAGE, sodium dodecyl sulfate polyacrylamide gel electrophoresis; UGM, urinary glycoprofile marker; CRPC, castration-resistant prostate cancer; BPH, benign prostate hyperplasia; mCRPC, metastatic castration-resistant prostate cancer; miRNA, microRNA; mRNA, messenger RNA; lncRNA, long noncoding RNA; AGR2, anterior gradient 2; PSEP, prostatic exosomal protein; LRG1, leucine-rich alpha-2-glycoprotein 1; ITIH3, inter-alpha-trypsin inhibitor heavy chain H3; NEU3, neuraminidase 3; PSA, prostate-specific antigen; PSMA, prostate-specific membrane antigen; AR-V7, androgen receptor variant 7.

**Table 2 ijms-26-00802-t002:** List of exosome contents with multiple biomarker roles in prostate cancer.

Year	Authors (et al.)	Exosome Contents	Type of Exosome Content	Detection Method	Role of Biomarkers in Disease	Sample Source	Sample Size	Change in Expression	Type of Study	Reference
2021	Li Z	miR-375, miR-451a, miR-486-3p, and miR-486-5p	miRNA	Next-generation sequencing	Diagnosis and prognosis	Urine	PC: 47, BPH: 29, HC: 25	miR-375: increased; miR-451a, miR-486-3p and miR-486-5p: reduced	Case–control	[57]
2021	Macías M	CCL2, CXCL5, S100A9 and TGF-ß	Cytokine	Magnetic bead-based immunoassay and ELISA	Diagnosis and treatment response	Blood	PC: 26, HC: 16	Increased	Case–control	[64]
2018	McKiernan J	ERG, PCA3, and SPDEF (ExoDx)	mRNA	qRT-PCR	Diagnosis and prognosis	Urine	503	Increased	Cohort	[67]
2016	McKiernan J	ERG, PCA3, and SPDEF (ExoDx)	mRNA	qRT-PCR	Diagnosis and prognosis	Urine	Training cohort: 255 Validation cohort: 519	Increased	Cohort	[68]
2022	Chu L	DHEA, DHEAS, testosterone, DHT	Hormone	Liquid chromatography tandem mass spectrometry	Diagnosis and prognosis	Urine	PC: 231, HC: 55	DHEA, DHEAS: increased; testosterone, DHT: reduced	Case–control	[69]
2022	Hansen EB	circSMARCA5, circHIPK3, circACVR2A, circN4BP2L2, and circMAN1A2	circRNA	Total RNA sequencing	Diagnosis and prognosis	Blood	LPC: 21, MPC: 6, HC: 27	Increased	Case–control	[126]
2023	Wei C	PSA	Protein	ELISA	Diagnosis and prognosis	Urine	272 participants undergoing prostate biopsy	Increased	Cohort	[127]
2021	Salvi S	CD62P, CD41b, CD42a, CD29, CD31, CD9, CD63, and CD24	Protein	Flow cytometry	Diagnosis and prognosis	Blood and urine	PC: 10, BPH: 10, healthy donors: 10	Reduced	Case–control	[128]
2024	Matijašević Joković S	PSMA and Caveolin-1	Protein	Western blot	Diagnosis and prognosis	Blood	PC: 39, BPH: 33	Increased	Case–control	[129]
2023	Tao W	AC015987.1, CTD-2589M5.4, RP11-363E6.3	lncRNA	RNA sequencing	Diagnosis and prognosis	Urine	Training cohort: 350 Validation cohort: 232,251	Increased	Cohort	[130]
2012	Bryant RJ	miR-141, miR-375, miR-107, and miR-574-3p	miRNA	qRT-PCR	Diagnosis and prognosis	Blood	PC: 78, HC: 28	Increased	Case–control	[131]
2021	Albino D	miR-424	miRNA	RT-qPCR	Diagnosis and prognosis	Blood	BPH: 6, primary PC: 25, mCSPC: 16, mCRPC: 17	Increased	Case–control	[132]
2016	Alhasan AH	miR-200c, miR-605, miR-135a, miR-433, and miR-106a	miRNA	qRT-PCR	Diagnosis and prognosis	Blood	VHR PC: 9, LR PC: 9, healthy donors: 10	miR-200c, miR-605, miR-135a: reduced; miR-433 and miR-106a: increased	Case–control	[133]
2021	Khanna K	STEAP1	Protein	Western blot	Diagnosis and prognosis	Blood	PC: 121. HC: 55	Increased	Case–control	[134]
2019	Bryzgunova OE	miR-30a: miR-125b; miR-425: miR-331; miR-29b: miR-21; miR-191: miR-200a; miR-331: miR-106b	miRNA	qRT-PCR	Diagnosis and treatment response	Urine	PC: 10, HC: 10, BPH: 10	Increased	Case–control	[135]
2020	Wang Y	miR-181a-5p	miRNA	Deep sequencing and microRNA chip array	Diagnosis and prognosis	Blood	nbmPCa: 35, BPH: 23, bmPCa: 16	Increased	Case–control	[136]
2018	Li S	EphrinA2	Protein	ELISA and Western blot	Diagnosis and prognosis	Blood	PC: 50, BPH: 21, HC: 20	Increased	Case–control	[137]
2020	Wang WW	sncRNA-Sentinel™	sncRNA	qRT-PCR	Diagnosis and prognosis	Urine	Test cohort: 235 Validation cohort: 613,823	Increased	Cohort	[138]
2018	Bhagirath D	miRNA-1246	miRNA	qRT-PCR	Diagnosis and prognosis	Blood	PC: 44, BPH: 4, HC: 8	Increased	Case–control	[139]
2016	Park YH	PSMA	Protein	ELISA	Diagnosis and prognosis	Blood	PC: 82, BPH: 28	Increased	Case–control	[140]
2024	Pang B	LAMB1 and Histone H4	Protein	ELISA and Western blot	Diagnosis and prognosis	Blood and urine	HC: 15, LPC: 30, MPC: 15	Increased	Case–control	[141]
2023	Lei Y	miR-222, miR-1290, miR-182, miR-21, miR-221, and miR-10b	miRNA	Dual-surface-protein-guided miRNA profiling	Diagnosis and prognosis	Blood	PC: 47, HC: 27	Increased	Case–control	[142]
2021	Ali HEA	miR-6068, miR-1915-3p, miR-3692-3p, miR-3939, miR-6716-5p, and miR-3692-3	miRNA	qRT-PCR	Diagnosis and prognosis	Blood	N: 150	Increased	Cohort	[143]
2023	Tao W	circCEP112, circFAM13A, circBRWD1, circVPS13C, and circMACROD2	circRNA	RNA sequencing	Prognosis and treatment response	Blood	Training cohort: 203 Validation cohort: 183,166	circCEP112, circFAM13A, and circBRWD1: increased; circVPS13C and circMACROD2: reduced in positive OS	Cohort	[144]
2024	Erdmann K	AMACR, PCA3, and PCAT29	mRNA	qRT-PCR	Prognosis and treatment response	Urine	N: 72	Increased	Cohort	[145]
2022	Wang C	AR-V7	mRNA	qRT-PCR	Prognosis and treatment response	Urine	mCRPC: 34 (ABI-Sta: 16, ABI-Res: 18), HC: 20	Increased	Case–control	[146]

PC, prostate cancer; HC, healthy control; LPC, localised prostate cancer; MPC, metastatic prostate cancer; CRPC, castration-resistant prostate cancer; BPH, benign prostate hyperplasia; ELISA, enzyme-linked immunosorbent assay; qRT-PCR, quantitative real-time polymerase chain reaction; RT-qPCR, reverse transcription quantitative polymerase chain reaction; OS, overall survival; DHEA, dehydroepiandrosterone; DHEAS, dehydroepiandrosteronesulfate; DHT, dihydrotestosterone; mCRPC, metastatic castration-resistant prostate cancer; mCSPC, metastatic castration-sensitive prostate cancer; bmPCa, bone metastatic prostate cancer; nbmPCa, non-bone metastatic prostate cancer; VHR PC, very-high-risk prostate cancer; LR PC, low-risk prostate cancer; ABI-sta, stable response to abiraterone; ABI-res, resistant to abiraterone; PSA, prostate-specific antigen; PSMA, prostate-specific membrane antigen; STEAP1, six-transmembrane epithelial antigen of the prostate 1; miRNA, microRNA; mRNA, messenger RNA; lncRNA, long noncoding RNA; sncRNA, small noncoding RNA; circRNA, circular RNA; AR-V7, androgen receptor variant 7.

### 3.3. Exosomes Predicting Disease Grade and Progression

The exosomal contents in liquid biopsy can be used to predict the prognosis of prostate cancer, like the grade of disease, metastasis, overall survival, and biochemical recurrence-free survival, as examined in the 48 articles reviewed (Table 3). miRNA is the most frequently studied exosomal load (18 articles), followed by proteins and gene expression based on mRNA, with 11 and 10 articles, respectively.

Forty-two different miRNAs were investigated and found to be prognostic markers for prostate cancer. Two articles found downregulation of miRNAs [56,146]. Most articles measured prognosis using the category of grade or disease severity, while some articles measured the projecting value of exosomal miRNA by predicting the metastasis [132,142,147,148]. Wang et al. studied a large cohort of prostate cancer patients and found three different sncRNAs—by using the platform Sentinel^TM^—have higher sensitivity and specificity in diagnosing and predicting high-grade prostate cancer [138].

Twelve different types of circRNAs have been investigated in four studies for their role in prostate cancer prognosis with significant statistical deviations from the norm [144,149,150,151,152]. Zavridou et al. studied the DNA methylation of the *GSTP1* and *RASSF1A* genes and found that they correlate with the OS [153]. Tao et al. found that specific lncRNAs (AC015987.1, CTD-2589M5.4, and RP11-363E6.3) can be potentially used for decision-making in active disease surveillance [130], while Kretschmer and co-workers utilised the ExoDx method in a large cohort of 2,066 patients for determining the prognostic value and found it can be used to classify grade group 1 to grade group 3 disease and can be used for active surveillance [154]. All the studies that studied the prognostic value of exosomal biomarkers found an increase in the expression level associated with poor progression or high-grade disease except for the study of Ruiz-Plazas et al., which showed the reduced expression of some miRNAs associated with high-risk disease (Table 3).

**Table 3 ijms-26-00802-t003:** List of exosomal biomarkers predicting prognosis of prostate cancer.

Year	Authors (et al.)	Exosome Contents	Type of Exosome Content	Detection Method	Sample Source	Sample Size	Change in Expression	Type of Study	Reference
2024	Lee J	miR-6880-5p	miRNA	Microarray and RT-qPCR	Blood	PC: 17, HC: 5	Reduced	Case–control	[155]
2021	Shin S	miR-21, miR-16, miR-142-3p, miR-451, miR-636	miRNA	RT-qPCR	Urine	PC: 149	Increased in poor survival	Cohort	[147]
2023	Temilola DO	miR-194-5p/miR-16-5p	miRNA	RNA sequencing	Blood	PC: 24, BPH: 10	Increased in metastasis	Case–control	[148]
2020	Li T	circ_0044516 (circRNA)	circRNA	Microarray	Blood	PC: 6, HC: 6	Increased in metastasis	Case–control	[149]
2023	Yang Z	circ-DHPS (circRNA)	circRNA	qRT-PCR	Blood	N: 31	Increased in metastasis	Cohort	[150]
2021	Zavridou M	GSTP1 and RASSF1A methylation	DNA methylation	Real-time methylation-specific PCR	Blood	mCRPC: 62, HC: 10	Increased in poor survival	Case–control	[153]
2022	Kretschmer A	PCA3, ERG, and SPDEF (ExoDx)	mRNA	qRT-PCR	Urine	N: 2,066	Increased in high risk disease	Cohort	[154]
2022	Zhu S	AKR1C3	mRNA	Digital droplet PCR	Blood	mCRPC: 62	Increased in poor survival	Cohort	[156]
2015	Donovan MJ	PCA3 and ERG	mRNA	RT-qPCR	Urine	N: 195	Increased in high grade disease	Cohort	[157]
2019	Del Re M	AR-V7	mRNA	Droplet digital PCR	Blood	N: 73	Increased in poor survival	Cohort	[158]
2019	Joncas FH	AR-V7	mRNA	Droplet digital PCR	Blood	PC: 89, HC: 10	Increased in poor survival	Case–control	[159]
2023	Wang JJ	ACP3, FOLH1, HOXB13, KLK2, KLK3, KLK4, MSMB, RLN1, SLC45A3, STEAP2, and TMPRSS2	mRNA	Reverse-transcription droplet digital PCR	Blood	MPC: 20, LPC: 20	Increased in metastasis	Cohort	[160]
2020	Ishizuya Y	Actinin-4	mRNA	qRT-PCR	Blood	MPC: 36	Increased in metastasis	Cohort	[161]
2024	Ding T	3-aminoquinoline, 6-dimethylaminopurine, diethyl-malonate, indole-3-acetic acid, n-3-hydroxypropyl phthalimide and n-benzoyl-2′-deoxycytidine	Metabolite	Liquid chromatography-electrospray ionisation tandem mass spectrometry	Urine	PC: 60, BPH: 40	Increased in high-grade disease	Case–control	[162]
2015	Huang X	miR-1290 and miR-375	miRNA	qRT-PCR	Blood	Screening cohort: 23 CRPC. Follow-up cohort: 100 CRPC	Increased in poor survival	Cohort	[163]
2023	Wang W	miR-222-3p	miRNA	RNA sequencing and qRT-PCR	Blood	ADPC: 15, CRPC: 15	Increased in high-grade disease	Cohort	[164]
2019	Fredsøe J	miR-151a-5p, miR-204-5p, miR-222-3p, miR-23b-3p, and miR-331-3p	miRNA	RT-qPCR	Urine	Cohort 1: 215 Cohort 2: 199 Cohort 3: 205	Increased in recurrence	Cohort	[165]
2022	Pudova EA	miRNA-148a-3p	miRNA	Next-generation sequencing and qRT-PCR	Blood	N: 11	Increased in high-grade disease	Cohort	[166]
2021	Ruiz-Plazas X	miR-221-3p, -222-3p and -31-5p, miR-193-3p, and -423-5p	miRNA	qRT-PCR	Urine and semen	N: 97	miR-221-3p, -222-3p and -31-5p: increased; miR-193-3p and -423-5p: reduced in high-risk disease	Cohort	[167]
2021	Bhagirath D	miR-28-5p and miR-148a-3p	miRNA	RNA sequencing and qRT-PCR	Blood	CRPC-adeno: 21, CRPC-NE: 6	Increased in high-grade disease	Cohort	[168]
2021	Kim MY	miR-26a-5p, miR-532-5p, and miR-99b-3p	miRNA	RT-qPCR	Urine	non-BCR: 49, BCR: 32	Increased in recurrence	Cohort	[169]
2020	Ye LF	PCA3 and PRAC	Protein	RT-qPCR	Urine	N: 89	Increased in high-risk disease	Cohort	[170]
2023	Gardani CFF	CD39	Protein	Flow cytometry	Blood	N: 25	Increased in poor prognosis	Cohort	[171]
2022	Lucien F	STEAP1, PSMA	Protein	Flow cytometry	Blood	N: 79	Increased in poor survival	Cohort	[172]
2021	Guo T	miR-423-3p	miRNA	RNA sequencing and RT-qPCR	Blood	Discovery cohort: Treatment-naive PC: 24, CRPC 2 Validation cohort 1: Treatment-naïve PC: 108, CRPC: 42 Validation cohort 2: Treatment-naïve: 30, CRPC: 30 Additional comparison: Non-CRPC patients on ADT: 36	Increased in high-grade disease	Cohort	[173]

PC, prostate cancer; HC, healthy control; CRPC, castration-resistant prostate cancer; BPH, benign prostate hyperplasia; qRT-PCR, quantitative real-time polymerase chain reaction; RT-qPCR, reverse transcription quantitative polymerase chain reaction; mCRPC, metastatic castration-resistant prostate cancer; MPC, metastatic prostate cancer; LPC, localised prostate cancer; ADPC, androgen-dependent prostate cancer; CRPC-adeno, castration-resistant prostate cancer with adenocarcinoma features; CRPC-NE, castration-resistant prostate cancer with neuroendocrine features; BCR, biochemical recurrence; miRNA, microRNA; mRNA, messenger RNA; circRNA, circular RNA; STEAP1, six-transmembrane epithelial antigen of the prostate 1; STEAP2, six-transmembrane epithelial antigen of the prostate 2; PSMA, prostate-specific membrane antigen; AR-V7, androgen receptor variant 7; ACTN4, actinin-4; ADT, androgen deprivation therapy.

### 3.4. Exosomes Related to Treatment Response

Out of twenty articles exploring the role of exosomes in different treatment responses, six examined gene expression, followed by miRNA and protein expression, at four articles each (Table 4). Among the proteins that were quantified, two different glycoproteins (P-glycoprotein and oncofetal glycoprotein-5T4) were found to be associated with docetaxel resistance and the presence of residual malignant cells, respectively [174,175]. In the study of Vardaki et al., the immune-checkpoint protein programmed death ligand 1 (PD-L1) was associated with a shorter OS with Radium-223 therapy [176].

Some studies evaluated the role of the exosomal content in determining the treatment response to chemotherapy agents, e.g., Docetaxel [177], while others examined their role in androgen receptor signalling inhibitors (ARSIs), e.g., Abiraterone [178]. In most studies, the biomarker expression is reduced with a favourable response, except one article that showed that urinary miRNA (miR-664a-5p) was significantly upregulated in patients responding to poly (ADP-ribose) polymerase (PARP) inhibitors [179].

Pukha et al. found that specific metabolites (glucuronate, D-ribose 5-phosphate, and isobutyryl-L-carnitine) can be used as a marker for successful prostatectomy [180]. Another study by Macías et al. reported that lncRNAs (CCL2, CXCL5, and S100A9) can predict the efficacy of surgical interventions and recovery [64]. Three studies showed that *androgen receptor splice variant 7* (*AR-V7*) gene expression can significantly predict the treatment response to ARSIs and hormone therapy [152,181,182]. Finally, a study by Malla and co-workers studied the role of miRNAs (let-7a-5p and miR-21-5p) in response to radiotherapy and found that their expression was elevated in high-risk prostate cancer patients compared to the intermediate-risk group [183].

**Table 4 ijms-26-00802-t004:** List of exosomal biomarkers predicting treatment response of prostate cancer.

Year	Authors (et al.)	Exosome Contents	Type of Exosome Content	Detection Method	Sample Source	Sample Size	Change in Expression	Type of Study	Reference
2017	Del Re M	AR-V7	mRNA	Digital droplet PCR	Blood	CRPC: 36	Increased in poor response to hormonal therapy	Cohort	[152]
2015	Kato T	P-glycoprotein (P-gp)	Glycoprotein	Western blot	Blood	Therapy-naïve patients: 6, TR PC: 4	Increased in Docetaxel resistance	Cohort	[174]
2009	Mitchell PJ	PSA, PSMA, oncofetal glycoprotein-5T4	Protein and glycoprotein	Electrophoresis and immuno-blotting	Urine	PC: 10, HC: 10	Reduced in ADT response	Case–control	[175]
2021	Vardaki I	PD-L1	mRNA	RNA sequencing	Blood	N: 25	Increased in poor response to radiotherapy	Cohort	[176]
2023	Jiang X	lincROR	lncRNA	qRT-PCR	Blood	MPC: 27	Increased in Docetaxel resistance	Cohort	[177]
2023	Kato T	H19	lncRNA	RNA-sequencing and digital droplet PCR	Blood	LPC: 58, MPC: 14, ARAT-naïve: 7, ARAT-resistant CRPC: 6	Increased in ARTA resistance	Cohort	[178]
2024	Kim MY	miR-664a-5p	miRNA	RNA sequencing and RT-qPCR	Urine	N: 8	Increased in PARP inhibitor response	Cohort	[179]
2017	Puhka M	Glucuronate, D-ribose 5-phosphate and Isobutyryl-L-carnitine	Metabolite	Liquid chromatography-tandem mass spectrometry	Urine	PC: 3, HC: 3	Reduced in prostatectomy	Case–control	[180]
2021	Del Re M	AR-V7	mRNA	Digital droplet PCR	Blood	mCRPC: 84	Reduced in ARTA response	Cohort	[181]
2019	Strati A	AR-V7, AR-567es	mRNA	RT-qPCR	Blood	mCRPC: 62, HC: 10	Increased in ARAT resistance	Case–control	[182]
2018	Malla B	let-7a-5p, miR-21-5p	miRNA	qRT-PCR	Blood	N: 25	Increased in radiotherapy resistance	Cohort	[183]
2023	Vardaki I	Stathmin-1 and ITSN1	mRNA	Transcriptome microarray	Blood	CRPC: 19	Increased in resistance to Cabazitaxel	Cohort	[184]
2021	Zhu S	TUBB3	mRNA	Digital droplet PCR	Blood	mCRPC: 52	Increased in poor Abiraterone response	Cohort	[185]
2024	Shutko EV	miR-125b, miR-660, miR-200b, miR-30e, and miR-375	miRNA	RT-qPCR	Urine	PC: 22, HC: 18	Reduced in radical prostatectomy response	Case–control	[186]
2015	Kharaziha P	MDR-1/3 and PABP4	Protein	Western blot	Blood	N: 6 (TR: 3, TS: 3)	Increased in Docetaxel resistance	Cohort	[187]

PC, prostate cancer; HC, healthy control; CRPC, castration-resistant prostate cancer; MPC: metastatic prostate cancer; qRT-PCR, quantitative real-time polymerase chain reaction; RT-qPCR, reverse transcription quantitative polymerase chain reaction; ADT, androgen deprivation therapy; ARTA, androgen receptor-targeted agent; PARP, poly ADP ribose polymerase; LPC: localised prostate cancer; BPH, benign prostate hyperplasia; mCRPC, metastatic castration-resistant prostate cancer; ARAT, androgen receptor axis-targeted therapy; PSA, prostate-specific antigen; PSMA, prostate-specific membrane antigen; PD-L1, programmed death ligand 1; AR-V7, androgen receptor variant 7; miRNA, microRNA; mRNA, messenger RNA; circRNA, circular RNA; lncRNA, long noncoding RNA; TR, docetaxel-reistant; TS, docetaxel-sensitive.

## 4. Discussion

This systematic review, encompassing 137 articles on the role of various exosomal loads in prostate cancer diagnosis, prognosis, and treatment response, highlights the significant advancements of noninvasive liquid biopsy in detection and making clinical decisions. The findings presented herein showcase the growing body of evidence supporting the utility of exosomes as potential liquid biopsy biomarkers for prostate cancer, particularly exosomal miRNAs and other loads.

### 4.1. Validity of the Study Design in Biomarker Study

Most studies included in the review were case–control studies (91 out of 137), indicating a robust interest in comparing exosomal biomarkers between prostate cancer patients and various control groups, including healthy individuals and those with BPH. A substantial number of cohort studies (46 articles), some of which included training and validation cohorts, further strengthened their analysis of the predictive exosomal biomarkers of prostate cancer in larger populations. These study designs are crucial for establishing the specificity and sensitivity of potential biomarkers.

### 4.2. Variation in Liquid Biopsy Samples

Trujillo et al. discussed the limitations of traditional tissue biopsies in clinical settings and the need for noninvasive alternatives like liquid biopsies [188]. The diversity of biological samples used for exosome isolation, primarily blood and urine, indicates the versatility of liquid biopsies. The utilisation of liquid biopsies to capture the dynamic nature of prostate cancer is a significant advancement, as traditional methods fall short in determining tumour heterogeneity and evolution [188]. The limited exploration of other fluids, such as semen and saliva, suggests that further research is needed to understand the full potential of the use of exosomal cargo for the diagnosis of prostate cancer. The variety of liquid biopsy sources indeed opens the possibility of developing a wide range of assays and options for patients with prostate cancer.

### 4.3. Heterogeneity in Biomolecule Detection Method

The detection methods used by different research groups for similar biomolecules differed from each other, thus creating the possibility of a non-harmonious conclusion. Amidst the heterogeneity of the assay methods, some studies validated certain detection methods (i.e., ExoDx) in large cohorts that are utilised commercially by patients and clinicians. Some studies utilised a next-generation sequencing platform for profiling RNA, which clearly is a more sensitive and high-throughput method of identifying a pool of deranged RNAs.

### 4.4. Emergence of miRNA as a New Exosomal Biomarker

A key finding of this review is that exosomal cargo could provide critical information to assist clinical decision-making, with miRNA, a type of noncoding RNA, being the most widely characterised exosomal load, with 49 articles focusing on its potential as a diagnostic, prognostic, and treatment response marker. The diagnostic efficacy of miRNA, often measured using ROC analysis, has shown promising results, with several studies reporting AUC values significantly higher than those of standard care testing (i.e., PSA). This notion aligns with the conclusion drawn by Jain et al. that urinary exosomal miRNAs might be highly instructive noninvasive biomarkers for early prostate cancer detection [189]. Additionally, Wang et al. noted that exosomal miRNAs could serve as reliable biomarkers for monitoring disease progression and treatment response [44].

The combined analysis of multiple biomarkers with their higher-level interactions, including miRNA and proteins, may offer a more comprehensive approach to diagnosis and prognosis, potentially leading to improved patient stratification and personalised treatment strategies. Combining the emerging miRNA biomarkers with traditional biomarkers like PSA has been suggested by Gaglani et al. to further enhance the diagnostic accuracy and staging of prostate cancer by integrating multiple biomarkers into tests [190].

### 4.5. Potential of Exosomes as Prognostic Biomarkers

In terms of prognosis, the review highlights that exosomal content can predict various outcomes, including the disease grade, metastasis, and OS. Specific miRNAs and lncRNAs as prognostic markers are particularly compelling, as they significantly correlated with the aggressiveness of the disease and the likelihood of recurrence. This notion agrees with the conclusion drawn by Gao et al., who emphasised the role of exosomal miRNA in monitoring prostate cancer invasion and metastasis [191]. The ability to stratify patients based on Gleason scores and predict biochemical recurrence using exosomal biomarkers could revolutionise active surveillance strategies, allowing for the more tailored management of prostate cancer [44].

### 4.6. A New Horizon of Assessing Therapy Response Using Exosomes

The use of exosome load analysis in assessing the treatment response is another intriguing aspect of this review. Identifying specific exosomal markers associated with resistance to therapies, such as Docetaxel and ARSIs, exhibited the potential of exosomes to inform treatment decisions, in line with Lorenc et al.’s forecast of the utility of exosomal biomarkers in predicting therapeutic responses and guiding therapeutic schemes [192]. Finally, the ability to monitor the treatment response through noninvasive methods could significantly improve long-term interventions, improving cancer management and allowing for treatment flexibility based on real-time longitudinal data [38,187].

### 4.7. Limitations, Challenges, and Prospects

Despite the promising findings, several limitations of the studies must be acknowledged. The predominance of retrospective studies raised concerns about potential biases, confounding factors, and the quality of data collected. The retrospective approach also limited the ability to draw definitive conclusions about the causality of biomarkers.

Most of the studies utilised a targeted detection method (i.e., qPCR or ELISA), thus limiting their ability to exclude relevant confounding biomolecules. A comprehensive biomolecule profiling (i.e., next-generation sequencing) and bioinformatics study is a prerequisite for optimizing the identification and validation of target biomolecules. A handful of studies further validated their finding in in vivo animal models, which, again, can be regarded as a limitation of most of the studies.

The lack of standardisation in the methods of exosome isolation and characterisation adds to the variability among the different studies, hence setting major obstacles for clinical translation. The standardisation of exosome isolation, quantification, and downstream analyses will be necessary to ensure reproducibility and comparability among studies.

The studies in this systematic review demonstrated marked heterogeneity regarding the selection of study subjects. This challenges any synthesis of results, reducing the ability to formulate definitive conclusions. For instance, using BPH patients as a control group may show variation in the baseline characteristics between control populations. There is a particular need for tailored studies addressing specific clinical scenarios, for example, distinguishing aggressive versus indolent prostate cancer, to address these gaps and further enhance the clinical applicability of exosomal biomarkers.

Finally, challenges remain in the integration of exosomal analysis into clinical practice and therapeutic workflows. Initially, the liquid biopsy of exosomal biomolecules may serve as complementary companions to standard-of-care investigations in achieving accuracy in the early detection of prostate cancer with reduced false positivity. In addition, exosomal components could provide actionable insights on disease progression and treatment resistance. In addition, effort is needed for developing user-friendly, point-of-care diagnostic devices based on exosomal biomarkers. The expertise of clinicians with an understanding of the biology and interpretation of exosomal biomarkers and their integration into existing diagnostic frameworks will be crucial for successful implementation.

## 5. Conclusions

In conclusion, this systematic review presents the latest and most comprehensive datasets demonstrating the transformative potential of exosomal liquid biopsy in diagnosis, prognosis, and treatment response in prostate cancer. Indeed, exosomal analysis holds much promise for improving the early detection and stratification of patients while reducing unnecessary invasive interventions. We strongly advocate for a multicentric consortium to conduct inclusive and comprehensive research to validate the emerging biomarkers in diverse population. We also emphasize the standardisation of exosome isolation and characterisation techniques and the development of a robust, comprehensive biomolecule profiling protocol to ensure effective biomarker discovery. With the arrival of machine learning and artificial intelligence-guided analytical tools, the multiplexing of various types of biomarkers with the analysis of their interactions may revolutionize the field, leading to breakthroughs in diagnosis and personalised therapy. Integrating exosomal analysis into routine clinical practice could lead to the successful translation of discussed biomolecules from the realm of research into clinical practice, paving the way for a new era of precision medicine in uro-oncology.

## Figures and Tables

**Figure 1 ijms-26-00802-f001:**
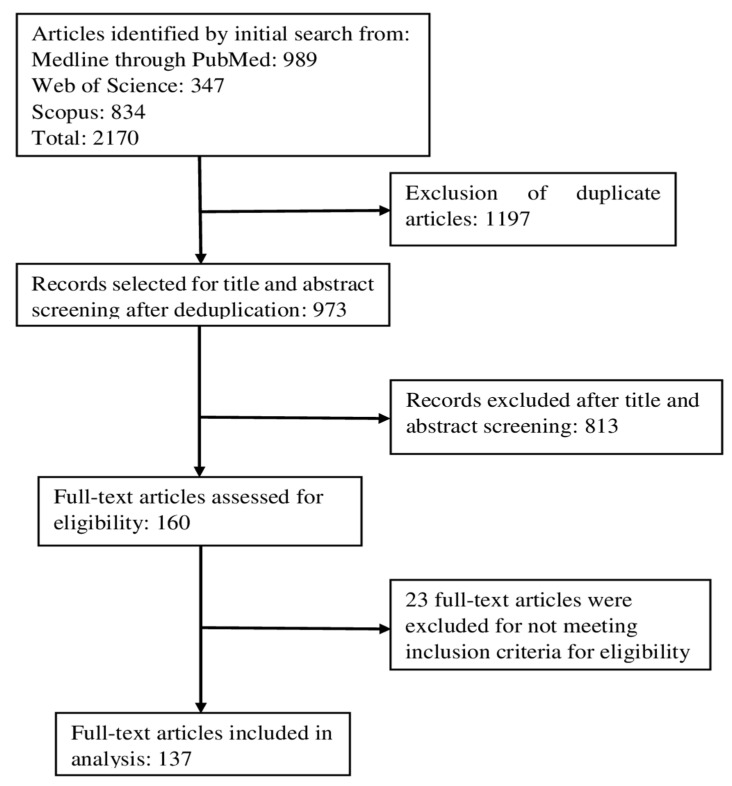
PRISMA flow diagram of the systematic review.

**Figure 2 ijms-26-00802-f002:**
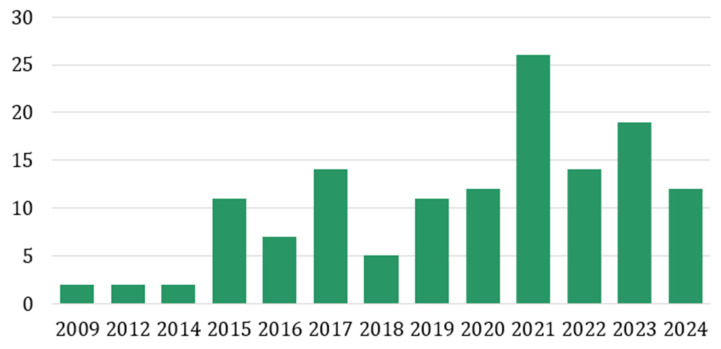
Distribution of selected articles based on year of publication.

**Figure 3 ijms-26-00802-f003:**
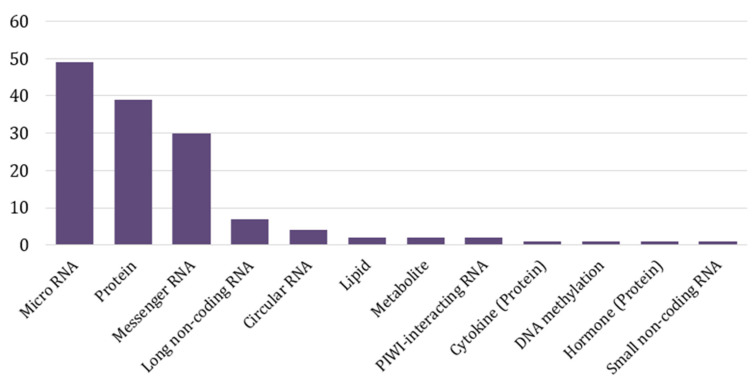
Bar chart of selected article numbers studying different exosomal components as prostate cancer liquid biopsy biomarkers.

**Figure 4 ijms-26-00802-f004:**
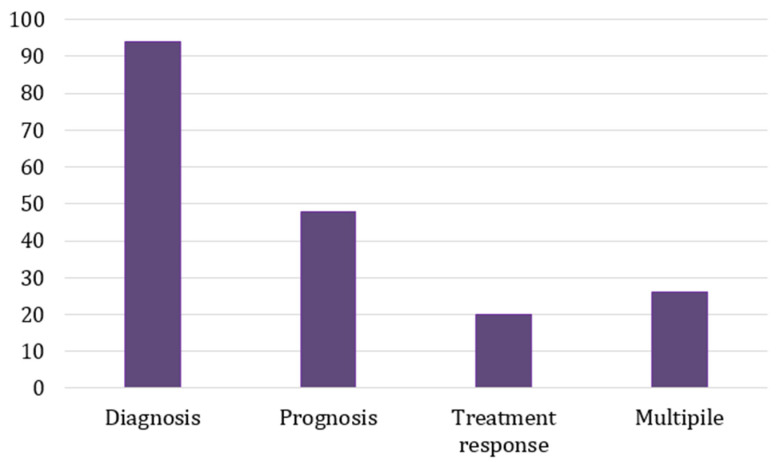
Bar chart of selected article numbers studying the particular role of exosomal components in prostate cancer.

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
