# Peer review of "Exosomal Liquid Biopsy in Prostate Cancer: A Systematic Review of Biomarkers for Diagnosis, Prognosis, and Treatment Response"

_ijms, 2025, doi:10.3390/ijms26020802_

Round 1

Reviewer 1 Report

Comments and Suggestions for Authors

·         The process to literature searching should be clearly defined in abstract.

·         The last paragraph in introduction should clearly state the research gap. All submitted reviews to IJMS should contain claims of novelty and give a justification for the publication of the paper. The work seems interesting only after reading the abstract. However, similar reviews have already been published on this topic (Am J Cancer Res. 2019 Jul 1;9(7):1309-1328; Ann Oncol. 2021 Feb 4;32(4):466–477; Cancers (Basel). 2022 Jul 4;14(13):3272; Biomedicines. 2022 Dec 2;10(12):3115; Int J Cancer. 2020 Nov 24;148(11):2640–2651; Crit Rev Oncol Hematol. 2020 Jan:145:102860 (Ref # 184); Front Cell Dev Biol. 2021 May 4;9:679527 (Ref # 187).  What will this systematic review accomplish in light of previous reviews?

·         In methods, provide more details on the search strategies, specifying the Boolean operators (AND, OR) and any additional filters used.

·         Statistical analysis should be described in sufficient details in a separate section. More details on methods for evaluating heterogeneity, use of subgroup (PCa patients and healthy individuals as control groups), and detecting of publication bias are needed.

·         Include a column in Tables 2 & 4 that represents the detection methods.

·         The discussion is way too short and very weak in its current form. The discussion should be prepared by including sub-sections according to detection methods, biomarkers, body fluids, and treatment options in liquid biopsies of PCa.

·         Provide more specific details about the limitations, such as the nature of the challenges and how they impacted the studies. Include specific examples or evidence for each limitation to strengthen the paper.

·         Clearly articulate the key findings and recommendations in conclusions.

Author Response

Dear Editor and Reviewer 1,

I am pleased to resubmit for publication the revised version of ijms-3387221 manuscript, entitled “Exosomal Liquid Biopsy in Prostate Cancer: A Systematic Review of Biomarkers for Diagnosis, Prognosis, and Treatment Response”.

Thankfully the reviewers provided us with a great deal of guidance, regarding how to better position the article. We are hopeful you agree that this revision will update our comprehensive review. All the comments have been addressed, as shown in the revised version of the manuscript, along with this point-by-point response to the reviewers' comments.

All corresponding are blue changes in the manuscript.

  • The process to literature searching should be clearly defined in abstract.

A sentence added in abstract (lines 29-32).

  • The last paragraph in introduction should clearly state the research gap. All submitted reviews to IJMS should contain claims of novelty and give a justification for the publication of the paper. The work seems interesting only after reading the abstract. However, similar reviews have already been published on this topic (Am J Cancer Res. 2019 Jul 1;9(7):1309-1328; Ann Oncol. 2021 Feb 4;32(4):466–477; Cancers (Basel). 2022 Jul 4;14(13):3272; Biomedicines. 2022 Dec 2;10(12):3115; Int J Cancer. 2020 Nov 24;148(11):2640–2651; Crit Rev Oncol Hematol. 2020 Jan:145:102860 (Ref # 184); Front Cell Dev Biol. 2021 May 4;9:679527 (Ref # 187).  What will this systematic review accomplish in light of previous reviews?

Research gaps in the available reviews are identified in last paragraph of introduction (lines 128-131).

  • In methods, provide more details on the search strategies, specifying the Boolean operators (AND, OR) and any additional filters used.

Search term with Boolean operators are mentioned in Method, first paragraph (lines 143-148).

  • Statistical analysis should be described in sufficient details in a separate section. More details on methods for evaluating heterogeneity, use of subgroup (PCa patients and healthy individuals as control groups), and detecting of publication bias are needed.

We did not perform meta-analysis as different exosomal contents were reported in this systematic review. Evaluating statistical heterogeneity (i.e. tree plot) and evaluating publication bias (i.e. funnel plot) would be applicable only if we performed a meta-analysis. However, we addressed the population heterogeneity of the studies in the discussion section as a limitation of this study and challenge of utilizing liquid biopsy as clinical application. Additionally, NOS was performed to assess risk of bias of the studies selected.

  • Include a column in Tables 2 & 4 that represents the detection methods.

The column mentioning detection method is added in all four tables.

  • The discussion is way too short and very weak in its current form. The discussion should be prepared by including sub-sections according to detection methods, biomarkers, body fluids, and treatment options in liquid biopsies of PCa.

Discussion section is restructured in subsections including the proposed subheadings.

  • Provide more specific details about the limitations, such as the nature of the challenges and how they impacted the studies. Include specific examples or evidence for each limitation to strengthen the paper.

Examples are provided in limitation subheading of discussion section (lines 433-465).

  • Clearly articulate the key findings and recommendations in conclusions.

Conclusion is reconstructed with articulation of findings and recommendation.

Reviewer 2 Report

Comments and Suggestions for Authors

The manuscript "Exosomal Liquid Biopsy in Prostate Cancer" provides a comprehensive review of exosomal biomarkers in prostate cancer diagnosis, prognosis, and treatment response. Analyzing data from 137 studies and over 17,000 patients, it highlights the potential of microRNAs (miRNAs) and other exosomal components to outperform traditional PSA testing in detecting prostate cancer and predicting disease progression. The paper's strength lies in its broad scope and focus on noninvasive diagnostics, addressing limitations in current screening methods.

However, the review also reveals challenges, including heterogeneous study designs and the lack of standardized exosome isolation protocols, which may hinder reproducibility. The reliance on retrospective data raises concerns about potential biases.

Overall, the manuscript is highly relevant to advancing precision medicine. Its call for standardization and multi-center studies underscores the need for further validation, positioning liquid biopsy as a promising tool in transforming prostate cancer care. 

To enhance the manuscript:

- consider expanding on the need for standardizing exosome isolation and analysis, as this is crucial for reproducibility and clinical adoption.

- address study heterogeneity with suggestions for harmonized protocols or tailored, larger prospective studies.

- highlight pathways for integrating exosomal biomarkers into existing clinical workflows.

- The conclusion could be made more impactful by emphasizing the potential of liquid biopsy to improve early detection and reduce unnecessary interventions.

Author Response

Dear Editor and Reviewers,

I am pleased to resubmit for publication the revised version of ijms-3387221 manuscript, entitled “Exosomal Liquid Biopsy in Prostate Cancer: Diagnostic, Prognostic, and Treatment Response Biomarkers”.

Thankfully the reviewers provided us with a great deal of guidance, regarding how to better position the article. We are hopeful you agree that this revision will update our comprehensive review. All the comments have been addressed, as shown in the revised version of the manuscript, along with this point-by-point response to the reviewers' comments.

All corresponding are blue changes in the manuscript.

The manuscript "Exosomal Liquid Biopsy in Prostate Cancer" provides a comprehensive review of exosomal biomarkers in prostate cancer diagnosis, prognosis, and treatment response. Analyzing data from 137 studies and over 17,000 patients, it highlights the potential of microRNAs (miRNAs) and other exosomal components to outperform traditional PSA testing in detecting prostate cancer and predicting disease progression. The paper's strength lies in its broad scope and focus on noninvasive diagnostics, addressing limitations in current screening methods.

However, the review also reveals challenges, including heterogeneous study designs and the lack of standardized exosome isolation protocols, which may hinder reproducibility. The reliance on retrospective data raises concerns about potential biases.

Overall, the manuscript is highly relevant to advancing precision medicine. Its call for standardization and multi-center studies underscores the need for further validation, positioning liquid biopsy as a promising tool in transforming prostate cancer care.

Thank you for your positive reinforcement and constructive feedback. We appreciate the opportunity to revise our work for consideration for publication.

To enhance the manuscript:

  • consider expanding on the need for standardizing exosome isolation and analysis, as this is crucial for reproducibility and clinical adoption.

The issue is addressed in third paragraph of discussion section 4.7 (lines 445-448).

  • address study heterogeneity with suggestions for harmonized protocols or tailored, larger prospective studies.

Addressed in fourth paragraph of discussion section 4.7 (lines 449-455).

  • highlight pathways for integrating exosomal biomarkers into existing clinical workflows.

Addressed in last paragraph of discussion section 4.7 (lines 456-465).

  • The conclusion could be made more impactful by emphasizing the potential of liquid biopsy to improve early detection and reduce unnecessary interventions.

Included in conclusion (lines 470-472 of conclusion section).

Round 2

Reviewer 1 Report

Comments and Suggestions for Authors

Dear Authors,

Unfortunately, the research gap is not clearly defined (comment 2). I would suggest to present the aim of the paper with regards to what is currently known by other reviews (Am J Cancer Res. 2019 Jul 1;9(7):1309-1328; Ann Oncol. 2021 Feb 4;32(4):466–477; Cancers (Basel). 2022 Jul 4;14(13):3272; Biomedicines. 2022 Dec 2;10(12):3115; Int J Cancer. 2020 Nov 24;148(11):2640–2651; Crit Rev Oncol Hematol. 2020 Jan:145:102860 (Ref # 184); therefore highlighting the added value of this review.

Author Response

Dear Editor and Reviewer 1,

I am pleased to resubmit for publication the revised version of ijms-3387221 manuscript, entitled “Exosomal Liquid Biopsy in Prostate Cancer: A Systematic Review of Biomarkers for Diagnosis, Prognosis, and Treatment Response”.

We appreciate you taking the time to provide your comment on the revised paper. Your feedback during the second minor-revision round has been addressed and is reflected in the final version of the manuscript. This corresponding change is highlighted in red within the manuscript.

The research gap has been clarified (lines 128–135), incorporating the references you recommended, which are now cited (references 40–45).